# Research on the Electron Structure and Antimicrobial Properties of Mandelic Acid and Its Alkali Metal Salts

**DOI:** 10.3390/ijms24043078

**Published:** 2023-02-04

**Authors:** Renata Świsłocka, Grzegorz Świderski, Justyna Nasiłowska, Barbara Sokołowska, Adrian Wojtczak, Włodzimierz Lewandowski

**Affiliations:** 1Department of Chemistry Biology and Biotechnology, Bialystok University of Technology, Wiejska 45E, 15-351 Bialystok, Poland; 2Wacław Dabrowski Institute of Agricultural and Food Biotechnology—State Research Institute, Rakowiecka 36, 02-532 Warsaw, Poland

**Keywords:** mandelic acid, alkali metal, antimicrobial properties, molecular spectroscopy, electron structure

## Abstract

This article investigated the structure, and the spectroscopic and antimicrobial properties of mandelic acid and its alkali metal salts. The electron charge distribution and aromaticity in the analyzed molecules were investigated using molecular spectroscopy methods (FT-IR, FT-Raman, ^1^H NMR, and ^13^C NMR) and theoretical calculations (structure, NBO, HOMO, LUMO, energy descriptors, and theoretical IR and NMR spectra). The B3LYP/6-311++G(d,p) method was used in the calculations. The antimicrobial activities of mandelic acid and its salt were tested against six bacteria: Gram-positive *Listeria monocytogenes* ATCC 13932, *Staphylococcus aureus* ATCC 25923, *Bacillus subtilis* ATCC 6633, and *Loigolactobacillus backii* KKP 3566; Gram-negative *Escherichia coli* ATCC 25922 and *Salmonella* Typhimurium ATCC 14028, as well as two yeast species, Rhodotorulla mucilaginosa KKP 3560 and Candida albicans ATCC 10231.

## 1. Introduction

Some of the aromatic acids of natural origin have antibacterial, anti-inflammatory, and anti-cancer properties. The best known of the acids that have been used as preservatives is benzoic acid [1,2]. In the food and cosmetic industry, alkali metal salts of these acids are usually used due to their better solubility, bioavailability, and increased biological potential [3,4]. An example is sodium benzoate. There are many compounds of natural origin with high biological activity. One of them is mandelic acid, which, due to its biological properties, has been used in the cosmetics and pharmaceutical industries [5,6,7]. Mandelic acid (MA) and its derivatives are an essential class of chemical compounds widely used in drug synthesis and stereochemistry research. In nature, the mandelic acid degradation pathway has been widely identified and analyzed as a representative pathway for the degradation of aromatic compounds [8]. MA and its derivatives are common substances, and their degradation pathways have been discovered in various Gram-negative and Gram-positive bacteria, filamentous fungi, and yeasts. It shows activity against, among others, Gram-positive bacteria (*Listeria monocytogenes* and *Staphylococcus aureus*) and Gram-negative bacteria (*Klebsiella pneumoniae* and *Pseudomonas aeruginosa*) [9,10]. (R)-mandelic acid (R-MA) is a precursor of semi-synthetic antibiotics penicillin and cephalosporins, and anti-obesity drugs [11]. (S)-mandelic acid (S-MA) is used for the synthesis of substituted cyclopentenones and commercial drugs, including non-steroidal anti-inflammatory drugs celecoxib and deracoxib [11], anticancer drugs, or anticoagulants [12]. Mandelic acid and its hydroxy derivatives have been quite well described in the literature [13,14,15]. However, there are few reports on mandelic acid salts and complexes with metals. The aim of this study was to compare the structure of alkali metal salts of mandelic acid and their antimicrobial properties to determine the potential uses of these compounds. The structure of chemical compounds affects their biological activity. Literature data and the results of previous studies have shown that alkali metals disturb the electron charge distribution in the aromatic ring of carboxylic acids [16,17,18,19]. The aromaticity of ligands, under the influence of salt formation, decreases in the Li–Na–K–Rb–Cs series with the increase in the metal ionic potential [20,21,22]. With the increasing destabilization of the aromatic system, the reactivity of the compounds increases. At the same time, it should be noted that alkali metal salts are usually more soluble and better absorbed than free ligands and thus show a higher biological activity [23]. Among the alkali metal salts with organic ligands, lithium, sodium, and potassium salts are mainly used in industry [24,25,26]. To study the electronic structure, we used spectroscopic methods (FT-IR, FT-Raman, NMR) and theoretical calculations of quantum chemistry. The antimicrobial properties of MA and its salts with alkali metals (Li, K, Na) were evaluated. The MIC and MBC parameters were determined [27]. Due to the natural ability of MA to chelate metal ions, and taking into account the benefits of alkali metals for human health, there is a need to test the properties of newly developed salts.

## 2. Results and Discussion

### 2.1. IR and Raman Spectroscopy

In the experimental FT-IR (KBr, ATR) and Raman spectra of mandelic acid, there are bands originating from the vibrations of the carboxyl and hydroxyl groups, as well as bands from the vibrations of the aromatic ring [28]. FT-IR spectra for mandelic acid and lithium, sodium, potassium, rubidium, and cesium salt of mandelic acid registered in KBr pellets, as presented in Figure 1. After the formation of the alkali metal salt with mandelic acid, characteristic changes in the ligand spectrum are observed (Table 1).

The spectra were made using two techniques of infrared spectroscopy, i.e., by pressing the sample with the matrix compound KBr (Figure 1) and the ATR technique (attenuated total reflection on the surface) (Appendix A). Both techniques allow us to obtain similar results; however, using an apparatus equipped with a zinc selenide crystal, we obtained ATR spectra, in which not all signals were clear (the spectra are included in the supplement). On the other hand, the comparison of the FTIR spectra of KBr and ATR made it easier to determine the bands originating from the vibrations of the hydroxyl group.

The FTIR spectra of the tested compounds show bands originating from the vibrations stretching the hydroxyl group. In the spectra recorded in the KBr matrix, they are present at wave numbers around 3200–3400 cm^−1^ and they are wide, which results from the formation of hydrogen bonds with water molecules absorbed by the substance. In the ATR spectra, the intensity of these bands is clearly weaker. The bands of vibrations stretching the hydroxyl group from the aliphatic chain of the molecule are present at wavenumbers 3550–3560 cm^−1^. These are quite sharp intense bands present in the FTIR spectra of KBr, usually poorly visible in ATR spectra.

On the spectra of mandelates, wide characteristic bands appear associated with the vibration of the carboxylate anion COO^−^. Bands corresponding to asymmetric stretching vibrations ν_as_COO^−^ in IR_KBr_ occur in the range of 1583–1607 cm^−1^, and in IR_ATR_ in the range of 1582–1606 cm^−1^. In the Raman spectrum (Appendix A) in the range of 1585–1587 cm^−1^. In the theoretical spectrum, the IR values are in the range of 1586–1622 cm^−1^. Bands corresponding to symmetrical stretching vibrations ν_s_COO^−^ in IR_KBr_ occur in the range of 1360–1402 cm^−1^, and in IR_ATR_, in the range of 1359–1402 cm cm^−1^. However, in the Raman spectrum, they occur in the range of 1360–1402 cm^−1^. In the theoretical spectrum, the values are in the range of 1346–1397 cm^−1^. Bands corresponding to symmetrical deformation vibrations in the plane β_s_COO^−^ in IR_KBr_ occur in the range of 868–871 cm^−1^, and in IR_ATR_, in the range of 860–868 cm^−1^. In the Raman spectrum, they occur in the range of 869–875 cm^−1^. In the theoretical spectrum, the values for infrared IR are in the range of 789–803 cm^−1^. Bands corresponding to deformation vibrations outside the γ_s_COO^-^ plane in IR_KBr_ occur in the range of 622–626 cm^−1^, and in IR_ATR_ in the range of 620–627 cm^−1^. In the Raman spectrum, they occur in the range of 618–619 cm^−1^. In the theoretical spectrum, the values are in the range of 614–620 cm^−1^. Bands corresponding to asymmetric deformation vibrations in the plane β_as_COO^-^ in IR_KBr_ occur in the range of 507–525 cm^−1^. In the Raman spectra, these bands occur in the range of 510–511 cm^−1^. The theoretical spectra range from 498 to 515 cm^−1^. The formation of alkali metal salts with aromatic carboxylic acids is accompanied by changes in the distribution of the electron charge in the aromatic ring of the ligand. The vibration bands of the aromatic system in the tested compounds were numbered as adopted by Versanyi [29]. We can observe changes in the number, position, and intensity of the bands of the aromatic system. The wavenumbers of the bands originating from the ν(CH) stretching vibrations marked with numbers 20a, 20b, and 7b shifted toward lower values in alkali metal salts compared to mandelic acid. Moreover, some of the bands originating from ν(CC) stretching vibrations shifted toward lower wavenumbers in mandelic acid salts (e.g., 19a). In mandelic acid and in its salts with alkali metals, deformation bands are observed: bending in the β(CH) ring plane, bending vibrations outside the γ(CH) ring plane, skeletal vibrations in the α(CCC) plane, and deformation bands outside the φ(CC) plane of the skeleton. For these bands, characteristic changes in the spectra of the salt as compared to the spectra of the ligand were also observed. Most of these bands shift toward smaller values of wavenumbers, e.g., 9a, 1, and 16b. The analysis of infrared spectra detected in the KBr matrix and with the ATR technique showed that the use of the KBr matrix had no significant effect on the band shifts in the spectra of the tested compounds.

The decrease in the wavenumber values of the bands of the aromatic system of the ligand, and the decrease in the intensity or the disappearance of these bands in the spectra of the ligand after salt formation is related to the decrease in the force constants of the bonds. This proves an increase in the disturbance of the charge distribution in the aromatic ring. The electron system in the ring of alkali metal salts of aromatic carboxylic acids is less stable than in the ligand.

### 2.2. NMR Study

#### 2.2.1. ^1^H NMR

The values of chemical shifts of ^1^H protons measured experimentally for mandelic acid and its salts with alkali metals, and theoretically calculated using the B3LYP/6-311++G(d,p) (DFT) method, are presented in Table 2. The values of chemical shifts of the aromatic ring protons in mandelic acid are in the range of 7.34 ppm to 7.45 ppm. The aliphatic chain proton of mandelic acid, designated H7a (Figure 2), gives a signal of 5.08 ppm. The signal of the proton of the hydroxyl group of the aliphatic chain, designated No. H7b is located at value 12.69. The theoretical values of the calculated chemical shifts give separate signals for the H7a proton (5.47 ppm) and the H7b proton (3.37 ppm). The chemical shift value of the H8 carboxyl proton is 12.69 ppm (exp.) and the theoretical value is 6.60 ppm. A significant difference in the values of chemical shifts calculated theoretically and experimentally determined for proton No. 8 and aliphatic proton no. H7b is probably due to the presence of hydrogen bonds in the dimers forming in the real structure [30]. Theoretical calculations were performed for mandelic acid monomer.

The spectra of alkali metal salts of mandelic acid do not show bands associated with chemical shifts of protons of the carboxyl group. After the formation of the alkali metal salt, an increase in the electron density around the protons of the aromatic ring of mandelic acid is observed. In alkali metal salts, the electron density around the aromatic protons (H2, H3, H4, H5, and H6) increases, which can be observed in the form of a decreasing value of the chemical shift of these protons in the ^1^HNMR spectra of the salt with regard to the ligand spectrum. The increase in the electron density around protons H2 and H6 occurs regularly in the studied series in the direction of MA–Li–Na–K–Rb–Cs along with the increasing ionic radius of the metal forming the salt with the ligand. In the case of protons H3, H4, and H5, the trends of changes in the electron density in the tested series of alkali metal salts are similar, and a slight deviation from this rule is observed only in the rubidium salt. The decrease in the value of chemical shifts of the protons of the aromatic ring indicates an increase in the disturbance of the electronic system in the ligand structure—the aromaticity of the compounds in the tested series of salts decreases in the MA–Li–Na–K–Rb–Cs.

#### 2.2.2. ^13^C NMR

The values of chemical shifts of ^13^C carbons measured experimentally for mandelic acid and its salts with alkali metals and the theoretically calculated values using the DFT method are presented in Table 3. The chemical shifts of the carbon atoms in the aromatic ring marked with numbers C2 and C6 (Figure 2) adopt the same values as C3 and C5. In the tested series of metal salts, an increase in the electron density around the atoms of aromatic carbons C2, C3, C4, C5, and C6 was observed in the series MA–Li–Na–K–Cs. The location of the signals of these carbons in the ^13^CNMR spectra of the tested salts is shifted toward lower values. The values of chemical shifts of the carbons of the ^13^C aromatic ring in the spectra of mandelic acid salts change systematically with the change in the ionic potential of the salt-forming metal, except for rubidium, which does not follow these trends. In the case of the C1 carbon atom, a decrease in the electron density around this atom was also observed in the Ma–Li–Na–K–Cs series. Changes in the electron density around the aliphatic carbon atoms in the tested salts in relation to the acid do not show such characteristic tendencies with the change of the salt-forming metal. The electron density at the C7 carbon decreases in the tested salts relative to the acid (increase in the value of chemical shifts of signals). In the case of the carbon of the C8 carboxyl group, the electron density around this atom is higher in lithium and cesium salts, and lower in other salts. Similar changes in the chemical shift values were observed for the 13C NMR spectra calculated theoretically by the GIAO method (in DMSO solvent) for the optimized structures of mandelic acid and lithium, sodium, and potassium salts (calculated using the B3LYP/6-311++G(d,p)). The differences between the calculated and experimentally determined values of chemical shifts result from the lack of consideration of intermolecular interactions in theoretical models.

### 2.3. Theoretical Computations

#### 2.3.1. Structures and Aromaticity

Three structures of mandelic acid conformers were optimized by the DFT method. Figure 2 shows the optimized structures I–III and the monomer structure solved by X-ray diffraction [30]. Conformer II has the lowest energy. This structure is identical to the real structure of mandelic acid according to X-ray diffraction data (Figure 2). Further calculations of aromaticity, IR and NMR spectra, the energy of HOMO and LUMO orbitals, and NBO charge distribution were carried out for conformer II.

The influence of the formation of alkali metal salts on the π-electron system of the ligand (mandelic acid) was assessed using the calculated aromaticity indices. The degree of π-electron delocalization in a planar, cyclic system containing double bonds can be assessed using numerical values, assuming a value of 0 for the cyclic system of the cyclohexatriene-a non-aromatic compound and a value of 1 for the purely aromatic system benzene (or in the form of a value of 0–100). There are several criteria for evaluating aromaticity, including geometrical criteria, which describe aromaticity within the range of these values. An example is the HOMA index (harmonic oscillator model of aromaticity) and its components—the GEO geometric and EN energy contributions are based on the lengths of bonds in the ring [31]. In many of our studies on the physicochemical properties of complex compounds and metal salts with aromatic carboxylic acids, we used spectroscopic methods (FTIR, Raman, NMR, UV-VIS, and X-ray). These tools allowed us to assess, among other things, the influence of metals on the stabilization/destabilization of the π-electron system of the ligand. Calculated values of indices based on geometrical data (including HOMA) for theoretically modeled structures usually showed the same character of aromaticity changes as the experimental results obtained.

The HOMA index (harmonic oscillator model of aromaticity) differs from all other geometry-based ones by assuming another reference bond length. In this model, instead of the mean bond length, a concept of the optimal bond length is applied [31]:$$HOMA=1-\left[\alpha {\left(R_{opt}-R_{ar}\right)}^{2}+\frac{\alpha }{n}\;\sum^{\;}{\left(R_{ar}-R_{i}\right)}^{2}\right]=1-EN-GEO$$
where:

*R_opt_*—the optimal value of bond length. For the CC type of bond in the benzene ring, the R_opt_ value is equal to 1.334;

*R_i_*—length of the i^th^ bond;

*n*—number of bond lengths in the ring;

*R_ar_*—the average bond length;

*α*—normalization factor, necessary to obtain a HOMA value equal to 1 for ideally aromatic benzene or 0 for ideally alternating cyclohexatriene Kekulé ring.

Within the confines of the HOMA model, it is possible to obtain two components that describe different contributions to a decrease in aromaticity, i.e., (a) bond elongation (the EN component), and (b) bond length alternation (the GEO component). The value of the HOMA index is equal to 1 for the entire aromatic system; HOMA = 0 when the structure is non-aromatic and HOMA < 0 for the anti-aromatic ring.

Another index based on the geometry of the aromatic ring used to assess aromaticity is I6 (Bird’s Index). The value of Bird’s aromaticity index (I_5_, I_6_) describes the equation [32]:$$I=100\left\{1-\frac{V}{V_{k}}\right\}$$
where: V_k_ stands for the five-membered ring 35 and the six-membered 33.3, and V is calculated from the equation:$$V=\left(\frac{100}{n_{av}}\right){\left[\overset{n}{\underset{r=1}{\sum }}{\left(n_{r}-n_{av}\right)}^{2}/n\right]}^{\frac{1}{2}}$$
where: *n_av_*—average binding order, *n*—bond order based on bond length: n = (a/R)—b, a and b—parameter depending on the type of atoms in the bond.

The aromaticity of chemical compounds can also be assessed by magnetic criteria. Among several magnetic aromaticity criteria, the NICS (nucleus-independent chemical shift) index, defined as the negative value of the absolute chemical shift in the center of the ring, has gained great recognition. Aromatic compounds have negative values, while non-aromatic compounds have positive index values [33].

The values of the calculated aromaticity indices are presented in Table 4. Geometric indices (HOMA, GEO, EN, I6, ΔCC, and ΔCCC) were calculated on the basis of the length of bonds in the ring for structures modeled theoretically by the DFT method. NICS index values were calculated by the GIAO method for optimized structures. Geometric parameters ΔCC and ΔCCC, and the aromaticity indices of salts (lithium, sodium, and potassium) were compared to mandelic acid (conformer II, as the structure with the largest stability). A systematic increase in the parameters ΔCC and ΔCCC was observed in the series MA → MA–Li → MA–Na → MA–K, i.e., the diversity of bond lengths and angles increases. This is confirmed by the decreasing values of HOMA and I6 aromaticity indices in the same row. The change in the NICS value in the salts relative to the ligand also indicates that the aromaticity of mandelic acid decreases under the influence of salt formation.

The calculated values of the geometric indexes and the magnetic index NICS confirm the results of experimental studies, which showed that alkali metals disrupt the π-electron system of mandelic acid.

#### 2.3.2. Natural Bond Orbital (NBO)

For the theoretically optimized structures of mandelic acid molecules and lithium, sodium, and potassium mandelates, the electron charge distribution was calculated using the NBO method (Table 5). The numerical values of the electron charges on individual atoms are presented in Table 6. Under the influence of salt formation, the distribution of the electron charge in the mandelic acid molecule changes. The electron density around the C1 atom decreases in the series of Li–Na–K salts, while the electron density on the remaining carbon atoms of the aromatic ring increases in the calculated series of alkali metal salts with mandelic acid. Analogous changes in the electron charge distribution were observed in the experimental ^13^C NMR spectra recorded for alkali metal salts of mandelic acid. In the case of aromatic protons, a decrease in the electron density in alkali metal salts relative to mandelic acid was observed around the H2 and H6 protons, and an increase in the electron density around the H3, H4, and H5 protons. Experimental studies of ^1^H NMR showed that the electron density around all aromatic protons increases.

#### 2.3.3. Energy of HOMO and LUMO Orbitals and Reactivity Descriptors

For the optimized molecules, the shape and position of the HOMO and LUMO orbitals were determined (Figure 3). In the calculated molecules, the HOMO orbital is located on the π-electron region of the aromatic ring and the hydroxyl substituents. The LUMO orbital for the compounds in question includes mainly the carboxyl group, especially for salts of mandelic acid with alkali metals, where the orbital is located on the metal atom.

One of the current trends in chemical research is the prediction of the physicochemical and biological properties of a chemical compound from its structural parameters. The structure of a molecule determines its properties. Based on the theoretical energy values of HOMO/LUMO boundary orbitals, several important chemical descriptors were calculated (Table 6).

The reactiveness and stability of compounds can be predicted by assessing the difference value between HOMO orbital energy and LUMO orbital energy (∆). The higher the value of that difference, the lower the reactivity and stability of the compounds. On this basis, it was shown that the reactivity of the tested compounds increases in the series MA → MA–Li → MA–K → MA–Na (a decrease in aromaticity indices, i.e., HOMO Aj, was observed in the same series). The chemical potential and hardness of a molecule are also important descriptors of the overall molecule reactivity and charge transfer during a chemical reaction. Currently, a new theory applicable to the interpretation of bimolecular reactions, based on electrophilicity and nucleophilicity descriptors, is being intensively developed. The global electrophilicity index (ω) is assumed to take into account two tendencies within the molecule—on the one hand, the willingness to accept electrons from the other component of the reaction, and on the other, the “resistance” to donating one’s own valence electron. The electrophilicity scale currently in use was built by Domingo [34] based on (ω) indices calculated using the B3LYP functional in the functional base 6–31 g(d). He divided electrophiles into three groups: strong electrophiles (ω > 1.50 eV), moderate electrophiles (0.80 eV > ω >1.50 eV), and weak electrophiles (ω < 0.80 eV). According to this classification, the described compounds should be considered strong electrophiles.

#### 2.3.4. Electrostatic Potential Map

The electrostatic potential map shows the regions of the molecules related to their electrophilic (red) and nucleophilic (blue) reactivity (Figure 4). In mandelic acid, the hydroxyl group of the carboxylic moiety is susceptible to nucleophilic attack. The hydroxyl group of the aliphatic chain of mandelic acid is susceptible to electrophilic attack, as are the aromatic protons of the ligand. In the potassium salt of mandelic acid, a higher electrophilic susceptibility of the hydroxyl group and ring protons was observed than in lithium and sodium salt. The potassium salt shows increased reactivity compared to the other analyzed molecules.

### 2.4. Antimicrobial Study

Changes in the bacterial and yeast growth using Bioscreen C Pro are shown in Figure 5 and Figure 6. The bactericidal activity of MA was highly dependent on the MA concentration. The inhibitory effect of MA was observed for almost all the bacterial strains except *L. backii* (Figure 5 and Table 7). Inhibition of the growth of Gram-positive bacteria was noted in the presence of MA in the concentration of 1.50 mg/mL for *L. monocytogenes* and *B. subtilis*. The concentration required to inhibit *S. aureus* was 1.75 mg/mL. A bactericidal effect was observed for the abovementioned strains in the presence of MA in a concentration of 2.00 mg/mL beyond *B. subtilis*. Even 5 mg/mL MA was insufficient for this inactivation. In turn, the MIC value for Gram-negative bacterial strains was higher, and amounted to 2.00 mg/mL and 2.50 mg/mL for *E. coli* and *S*. *typhimurium*, respectively. MBCs were determined in the presence of MA in a concentration of 2.50 mg/mL. There was no significant effect of MA in a concentration below 5.00 mg/mL on the growing population of yeast strains. The elongation of the adaptive and logarithmic phases of the tested bacterial strains was observed with the increase in MA concentration. A few changes in the growth profiles were detected in the presence of MA at the higher concentrations (3.00 and 5.00 mg/mL) for both yeast strains. There was good agreement between the Gompertz model curves and data obtained by the Bioscreen C Pro device in most cases (Figure 7). The effect of mandelic acid on µ_max_ was calculated from the parameters B, D, C, and A obtained by fitting the Gompertz to the data collected from Bioscreen C Pro, as shown in Table 8. With the increase in MA concentration, µ_max_ significantly decreased in each strain. Moreover, similar concentrations of MA had a significantly different impact on the maximal growth rate of different strains (*p* < 0.05) (Table 9). Minimal or no inhibitory effect of the MA complex with alkali metals was observed on the tested microorganisms (Figure 5). The final concentration of bacterial cells, expressed as OD value, in the case of *L. monocytogenes* and *B. subtilis* was statistically higher in the presence of each of the MA salts than in MA. However, it was significantly lower than the concentration obtained for samples without any antimicrobial agent (control samples). In turn, the final concentration of *S. aureus, L. backii*, and *R. mucilaginosa* was also statistically higher in the presence of each of the MA salts than in MA. These OD values were closely associated with the results obtained for samples without any antimicrobial agent. Additionally, a higher OD value was observed for MA salts with Na and K metal ions than in control samples (for *L. backii,* and *R. mucilaginosa)*. This phenomenon was noted for *S. aureus*, *L. backii,* and *R. mucilaginosa*. The final concentration of *C.albicans* found close associations in the following samples: MA and its salts with Li and K ions and MA with Na and the control sample. The synthesis of MA with alkali metals probably caused the loss of antimicrobial properties even against bacterial strains. So far, the antibacterial activity of MA has been reported in a few publications [10,35,36]. Motamedifar et al. 2014. reported that both the MIC and MBC for methicillin-sensitive *S. aureus* strains of MA were 20 mg/mL. In turn, for methicillin-resistant strains, these parameters ranged from 20 to 40 mg MA/mL and 20–80 MA/mL, respectively. Fuursted et al. [35] have confirmed the antibacterial activity of MA against *Pseudomonas aeruginosa*, *Escherichia coli*, *Staphylococcus aureus*, *Klebsiella pneumoniae*, *Enterococcus faecalis*, *Enterobacter cloacae*, *Proteus mirabilis*, *Serratia marcescens,* and *Staphylococcus epidermidis*, albeit only as a mixture with lactic acid in the concentration of 500 mg/L. On the other hand, there was no inhibition of urinary tract pathogens e.g., *Proteus mirabilis* at concentrations above 5 mg/mL [37]. The influence of MA derivatives on microorganisms has been reported in relatively few publications. In most cases, the toxicity of MA ionic liquids was described against bacteria, yeast, and filamentous fungi. As an example, Prydderch et al. (2017) [37] described the toxicity of ten ionic liquids synthesized from renewable resource mandelic acid toward thirteen bacterial strains: *Staphylococcus aureus*, *Staphylococcus epidermidis*, *Escherichia coli*, *Klebsiella pneumoniae*, *Pseudomonas aeruginosa*, and *Enterococcus* sp., as well twelve fungal strains of the genera *Candida*, *Aspergillus*, *Trichosporon*, and *Absidia*. The authors showed that MA ionic liquids did not have high toxicity toward the bacterial strains screened against. In other findings, Ventura et al. [38] described the bacterial toxicity of imidazolium and pyridinium halide ionic liquids produced by a short synthesis from substituted mandelic acid derivatives. All tested substances had low toxicity toward *B. subtilis* and *E. coli*, *P. fluorescens*, and *P. putida* bacterial strains, although a significant range in IC_50_ values was obtained. Unfortunately, there are few data about MA and its salts with alkali metals.

## 3. Materials and Methods

### 3.1. Synthesis

A portion of mandelic acid (about 0.1 g) was dissolved in a solution of alkali metal hydroxide (LiOH, NaOH, KOH, RbOH, and CsOH) at a concentration of 0.1 M in a stoichiometric amount (1:1). The solution was heated to 50 °C and stirred for about 1 h. After the reaction, the mixture was filtered and slowly evaporated at room temperature. After the evaporation of water, crystalline substances were obtained, which were placed in an oven for 48 h at 120^°^C to get rid of the rest of the water. The synthesis was repeated three times, each time performing a preliminary FT-IR_ATR_ analysis of the salts obtained. Mandelic acid (98%) and alkaline metal hydroxides were purchased from Sigma Aldrich.

### 3.2. Spectroscopic Study

IR spectra were recorded using a BRUKER ALPHA spectrometer with a universal transmission adapter. Samples were prepared in a KBr matrix (200:1 ratio). The spectrum was measured in the range of 4000–400 cm^−1^ with a resolution of 4 cm^−1^. Infrared spectra were also recorded using the ATR technique using the BRUKER ALPHA spectrophotometer with an ATR attachment containing a zinc selenide (ZnSe) crystal. The measurement was made in the range of 4000–600 cm^−1^ with a resolution of 4 cm^−1^. Raman spectra were recorded using a BRUKER MultiRAM spectrophotometer in the range of 4000–400 cm^−1^. ^1^H NMR and ^13^C NMR spectra of the DMSO sample solution of the studied compound were recorded with a Bruker Avance II 400 MHz unit at room temperature with TMS as an internal reference.

### 3.3. Theoretical Computations

Optimization of the geometric structures of mandelic acid and its alkali metal salts was carried out using the B3LYP/6-311++G(d,p) method. The electron charge distribution was calculated for the optimal structures by the NBO (natural bond orbital) method. The energy of the HOMO and LUMO orbitals and the reactivity descriptors were calculated using the B3LYP/6-311++G(d,p) method. ^1^H NMR and ^13^C NMR spectra were calculated by the GIAO method using the DMSO solvent model. Calculations were made using the B3LYP/6-311++G(d,p) density functional method. For calculated IR spectra, the applied vibrational frequency scaling factors IR_factors_ = 0.98 [39]. All theoretical calculations were performed using the Gaussian 09 program package [40]. Based on the calculated bond lengths in the optimized structures, the HOMA, GEO, EN, I6, NICS, ΔCC, and ΔCCC aromaticity indices were calculated according to the equations described in the literature [31,32,33]. Maps of electrostatic potential distribution were made.

### 3.4. Antimicrobial Study

#### 3.4.1. Preparation of the Inoculum

The antimicrobial activities of mandelic acid and its three derivatives were tested against six bacteria: Gram-positive *Listeria monocytogenes* ATCC 13932, *Staphylococcus aureus* ATCC 25923, *Bacillus subtilis* ATCC 6633, and *Loigolactobacillus backii* KKP 3566; Gram-negative *Escherichia coli* ATCC 25922 and *Salmonella typhimurium* ATCC 14028, and two yeasts, *Rhodotorulla mucilaginosa* KKP 3560 and *Candida albicans* ATCC 10231.

The strains were stored in a frozen state before testing in 20% glycerol at −80 °C. Before each experiment, stock cultures were thawed at room temperature and streaked onto the appropriate agar medium. A single colony was transferred to 10 mL of sterile proper broth. Microorganisms were grown in a suitable medium and incubated in optimal conditions, as shown in Table 9. Bacteria were cultivated using Müeller–Hinton (MH) broth/agar (OXOID, Thermo Scientific, Hampshire, United Kingdom) or de Man–Rogosa–Sharpe (MRS) broth/agar (Merck, Darmstadt, Germany). Yeasts were cultivated using yeast extract–peptone–dextrose (YPD) broth/agar). Afterward, cultures were centrifuged (4000× *g*, 10 min, 4 °C) (Sorvall LYNX 6000 Centrifuge, Thermo Fisher Scientific, Massachusetts, United States). Sedimented cells were suspended in sterile physiological saline (SF). Then, inoculum suspensions with turbidity equivalent to 1 McFarland were prepared for each tested bacteria and yeast, which corresponded to an inoculum density in a range from 10^7^ to 10^8^ CFU/mL.

#### 3.4.2. Preparation of Almond Acid and Its Derivatives for Antimicrobial Activity Test

Mandelic acid (Sigma Aldrich, St. Louis, MI, USA) and its salts with alkali metals: lithium (Li), sodium (Na), and potassium (K) were used in this study. First, mandelic acid and its derivatives were dissolved in sterile distilled water to prepare stock solutions at ten times greater than the target concentration. As a result, the final concentrations of mandelic acid were 5.00; 3.00; 2.50; 2.25; 2.00; 1.75; 1.50; 1.25; 0.75 mg/mL; and 5.00 mg/mL for its salts with metals.

#### 3.4.3. Bioscreen C Pro Device

Antimicrobial activity against bacteria and yeast was investigated using the Bioscreen C Pro device (Yo AB Ltd., Growth Curves, Helsinki, Finland). The 100-well Honeycomb plate was filled with a liquid medium of 220 µL of MHB for bacteria and YPD broth for yeast, respectively, and supplemented with 30 µL of a mandelic acid solution or its salts in each well. The wells were inoculated with tested strains in the amount of 50 µL. Control samples possessed 50 µL of tested strains in 250 µL of broth without adding mandelic acid or its salts. Samples were incubated for 24 h at 37 °C or 48 h at 30 °C (bacteria) and 72 h at 25 °C (yeast) under steady-state conditions. Optical density (OD) was measured every 1 h with a 600 nm filter. The samples were shaken for 30 s before measurement. Each test was performed in triplicate.

#### 3.4.4. Minimum Inhibitory Concentration (MIC) and Minimal Bactericidal Concentration (MBC)/Minimal Fungicidal Concentration (MFC)

MIC is defined as the lowest concentration of an antibacterial agent that effectively inhibits the visible growth of microorganisms within a specified period under strictly controlled in vitro conditions. MBC is the lowest antibacterial agent concentration that destroys viable microorganism cells [27]. In this study, MICs were determined based on no observed OD increase during strain incubation in reference to the initial OD value. MBCs and MFCs were obtained from wells in which no turbidity was observed. For this purpose, a 20 µL droplet of the sample was inoculated on an appropriate agar medium and incubated in conditions dedicated to the tested microorganisms described above. The MA or its salt concentration at which no microbial growth was obtained was taken as the MBC.

#### 3.4.5. Growth Characteristics of the Tested Strains

The growth of microorganisms was described in concentrations of MA in which an inhibitory effect was not observed. The modified Gompertz model was used for that purpose, according to Equation (1). Gompertz curves and kinetics parameters were generated by fitting the Gompertz equation to the OD data obtained previously using the Bioscreen C Pro device. All curves were fitted, and parameters were derived using LabPlot 2.9.0 by KDE.
(1)$$L_{t}=A+Cexp\left[-exp\left(-B\left\{t-D\right\}\right)\right]$$
where:

OD(t)—optical density of the microorganisms’ cells at a wavelength of 600 nm;

t—time (h);

L(t)—is the OD at time t;

A—is the asymptotic OD value as t decreases indefinitely;

C—is the asymptotic amount of growth that occurs as t increases indefinitely;

B—is the relative growth rate at D;

D—is the time at which the absolute growth rate is at its maximum (h).

The maximum growth rate µ_max_ (2) was determined based on the Gompertz model.
(2)$$\mu_{max}=\frac{BC}{e}\;\left[h^{-1}\right]\;\;\;\;\;\;\;$$

#### 3.4.6. Statistical Analysis

Statistical analysis was performed using Statistica 14.0 (TIBCO Software, Palo Alto, CA, USA). The normality of the distribution was checked using the Shapiro–Wilk test. Equality of variance was checked using the Levene test and Brown–Forsythe test. To assess the significant differences in the maximum growth rate, a one-way analysis of variance (ANOVA) was performed. After checking the assumptions to show differences between the groups, Tukey’s HSD test was used (α = 0.05). Moreover, agglomerative clustering using single linkage was performed for results obtained for MA and its salts in a 5 mg/mL concentration. Dendrograms were used to represent hierarchical clustering for those results.

## 4. Conclusions

FT-IR, Raman, ^1^H NMR, and ^13^C NMR spectroscopic studies have shown that alkali metal ions forming salts with mandelic acid increase the disturbance of the electron system in the aromatic ring of the ligand. The degree of disturbance increases with the change of the metal ion in subsequent salts in accordance with the increasing ionic potential, i.e., in the series of Li–Na–K–Rb–Cs salts. Theoretical calculations carried out using the DFT method (B3LYP/6-311++G(d,p) for the optimized structures of mandelic acid and its salts with lithium, sodium, and potassium confirm the results of experimental studies. Calculations of aromaticity indices showed that the salts are characterized by lower aromaticity than mandelic acid. Along with the decrease in the aromaticity of the pi-electron system of the ligand ring under the influence of salt formation, the reactivity of the molecule increases, which was shown by calculations of the energy of the HOMO and LUMO orbitals, and reactivity descriptors calculated for the modeled structures. Obtained results confirmed the antibacterial activity of MA on the bacterial population in contrast with MA salts. Gram-negative bacteria showed greater resistance to MA than Gram-positive bacteria. The MIC value for bacterial strains differs depending on the species, while the MBC was the same for bacteria in which the composition of the cell wall was similar. However, the concentration of MA up to 5 mg/mL is insufficient to inhibit and kill *L. backii* and yeast cells. Therefore, more scientific studies need to be conducted to determine the antifungal activity of MA. The increase in the concentration of MA correlated with the decreasing value of the specific growth rate µmax. Cluster analysis has shown that the results obtained for most tested strains, control samples, and salts were very closely associated.

## Figures and Tables

**Figure 1 ijms-24-03078-f001:**
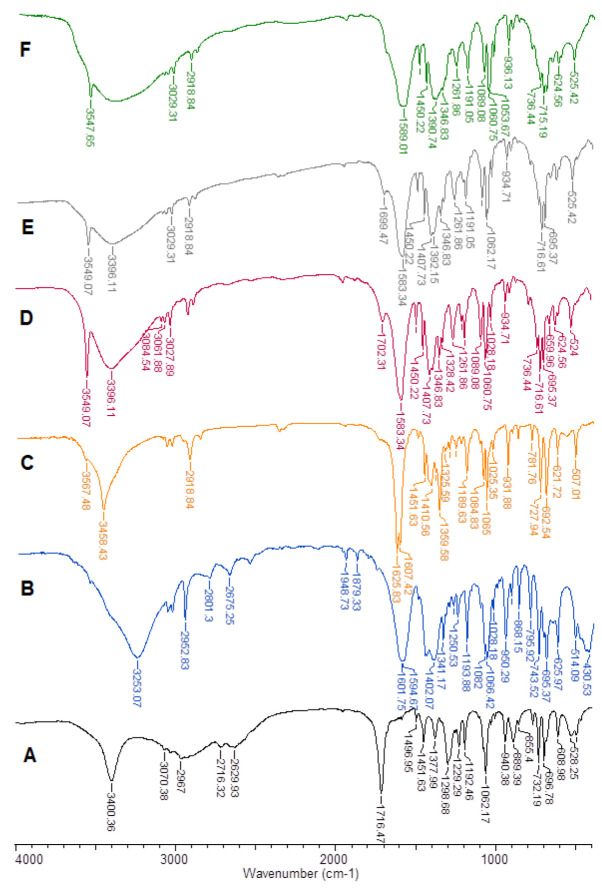
FT-IR spectra for mandelic acid (**A**) and lithium (**B**), sodium (**C**), potassium (**D**), rubidium (**E**), and cesium (**F**) salt of mandelic acid registered in KBr pellets.

**Figure 2 ijms-24-03078-f002:**
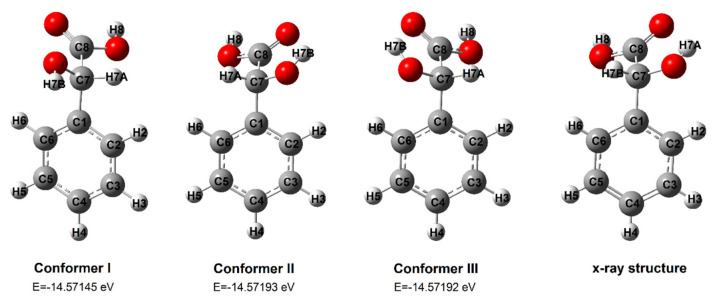
Conformers of mandelic acid (calculated in B3LYP/6-311++G(d,p)), and X-ray data structures [30].

**Figure 3 ijms-24-03078-f003:**
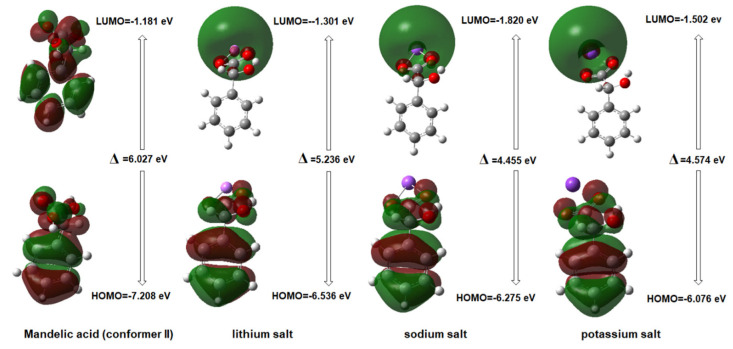
Shape and position of the HOMO and LUMO orbitals in the analyzed molecules.

**Figure 4 ijms-24-03078-f004:**
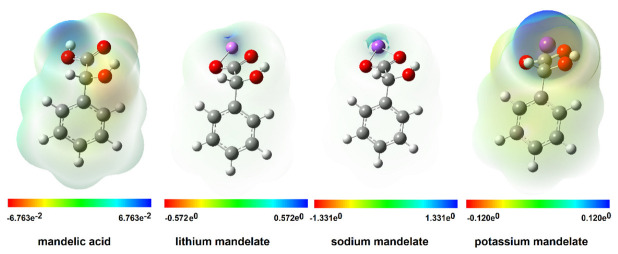
Electrostatic potential maps for mandelic acid and lithium, sodium, and potassium mandelate calculated in B3LYP/6-311++G(d,p).

**Figure 5 ijms-24-03078-f005:**
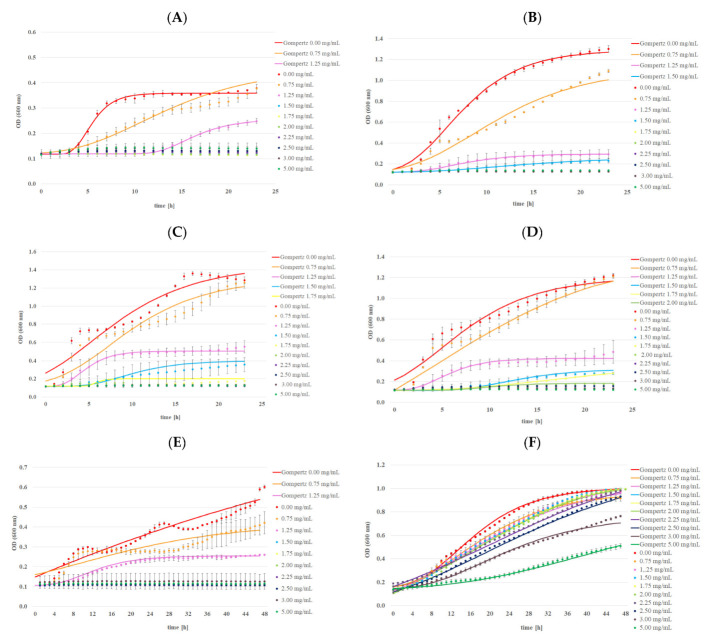
The effect of mandelic acid at different concentrations on the growth profiles of tested bacterial strains: (**A**) *L. monocytogenes* ATCC 13932, (**B**) *S. aureus* ATCC 25923, (**C**) *E. coli* ATCC 25922, (**D**) *S. typhimurium* ATCC 14028, (**E**) *B. subtilis* ATCC 6633, and (**F**) *L. backii* KKP 3566, using Bioscreen C Pro.

**Figure 6 ijms-24-03078-f006:**
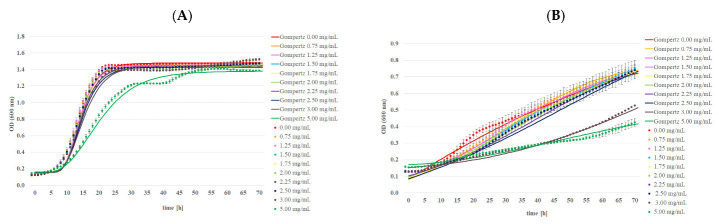
The effect of mandelic acid at different concentrations on the growth profiles of tested yeast strains: (**A**) *C. albicans* ATCC 10231 and (**B**) *R. mucilaginosa* KKP 3560, using Bioscreen C Pro.

**Figure 7 ijms-24-03078-f007:**
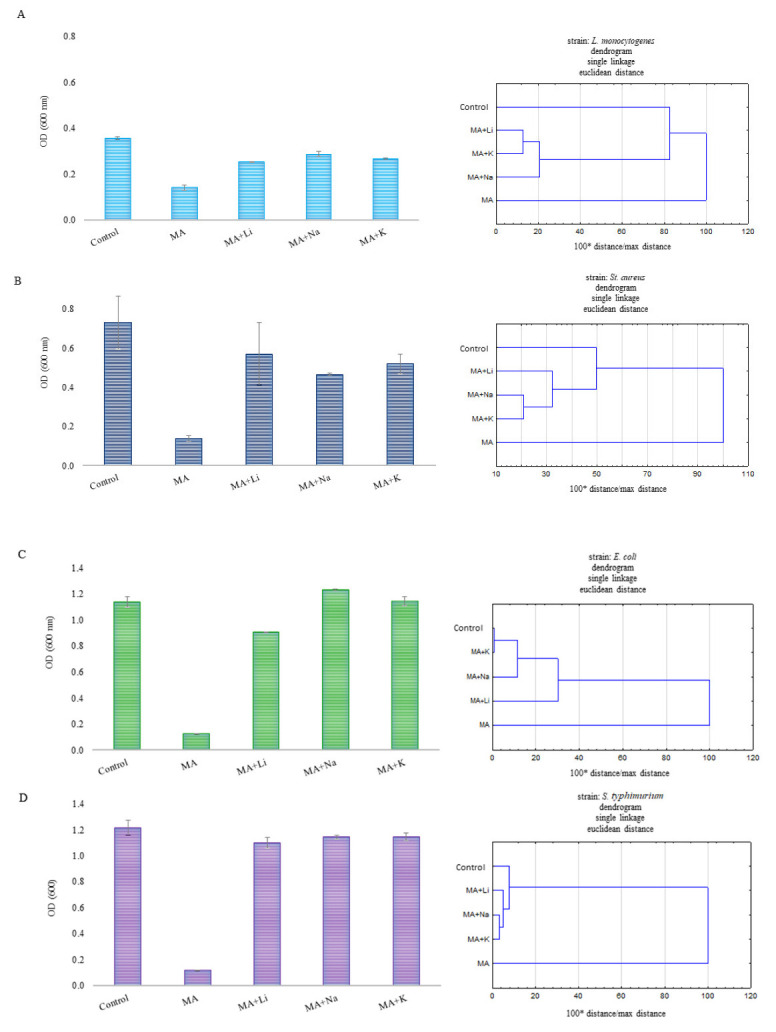
OD value on the last day of cultivation using Bioscreen C Pro (left) and dendrograms (right) for (**A**) *L. monocytogenes* ATCC 13932, (**B**) *S. aureus* ATCC 25923, (**C**) *E. coli* ATCC 25922, (**D**) *S. typhimurium* ATCC 14028, (**E**) *B. subtilis* ATCC 6633, (**F**) *L. backii* KKP 3566, (**G**) *C. albicans* ATCC 10231, and (**H**) *R. mucilaginosa* KKP 3560 in the presence of MA (5 mg/mL) and its salts (5 mg/mL). Control means the strain without any antimicrobial agent.

**Table 1 ijms-24-03078-t001:** The wavenumbers (cm^−1^) and assignments of bands from the FT-IR and FT-Raman spectra of lithium, sodium, potassium, rubidium, and cesium of mandelate.

Assignment [29]		MA–Li					MA–Na					MA–K					MA–Rb			MA–Cs	
		IR_KBr_	IR_ATR_	Raman	Teor.	Int.	IR_KBr_	IR_ATR_	Raman	Teor.	Int.	IR_KBr_	IR_ATR_	Raman	Teor.	Int.	IR_KBr_	IR_ATR_	Raman	IR_KBr_	IR_ATR_
		cm^−1^ (int.)	cm^−1^ (int.)	cm^−1^ (int.)	cm^−1^	KM/Mole	cm^−1^ (int.)	cm^−1^ (int.)	cm^−1^ (int.)	cm^−1^	KM/Mole	cm^−1^ (int.)	cm^−1^ (int.)	cm^−1^ (int.)	cm^−1^	KM/ Mole	cm^−1^ (int.)	cm^−1^ (int.)	cm^−1^ (int.)	cm^−1^	
	νOH_alif_	3549 w			3706	130.3	3567 m			3679	151.7	3549 s	3548 m		3653	167.5	3549s			3548 s	
	νOH(H_2_O)	3253 vs	3249 w				3458 s	3458 w				3396 s	3386 w				3396 vs	3376 w		3392 vs	3351 w
2	ν(CH)	3062 m	3064 vw	3059 s	3188	12.8	3062- w	3063 vw	3056 vs	3188	10.5	3062 m	3065 vw	3064 vs	3189	9.6	3063 s	3065 vw	3065 vs	3062 s	
20a	ν(CH)	3032 m	3038 vw		3179	22.3	3031 w	3034 vw		3178	26.0	3028 m	3028 vw		3177	29.6	3029 s	3031 vw		3029 s	
20b	ν(CH)	2953 s	2953 w	2954 m	3167	13.7	2956 vw			3164	17.7		2953 vw		3163	19.9		2950 vw	2950 w		
7b	ν(CH)				3157	0.1	2919 w	2916 vw	2918 m	3154	0.1	2919 m		2924 w	3153	0.1	2919 m			2919 m	
	νCHOH	2675 m			3019	18.2	2854 w			3018	21.3	2883 m			3012	24.2	2883 m			2883 m	
8a	ν(CC)				1641	5.9	1626 vs	1624 s		1641	9.9	1702 m			1641	14.1	1699 m				
	ν_as_COO; δCHOH	1602 vs	1596 vs	1587 w	1586	327.3	1607 vs	1606 s	1587 w	1606	361.8	1583 vs	1582 vs	1585 w	1622	1.6	1583 vs		1585 w	1589 vs	
19a	ν(CC)			1604 m					1603 m					1603 m	1616	323.3		1605 vs	1601 m		1609 vs
19b	ν(CC)	1497 m			1522	11.3	1496 w	1496 w		1522	12.9	1493 m	1493 w		1521	13.3	1493 m	1497 w		1493 m	1497 m
	βOH; δCHOH	1455 vs	1455 m		1479	28.0	1452 m	1452 m		1478	22.1	1450 m	1450 w		1478	21.1	1450 s	1455 w		1450 s	1456 m
14	ν(CC)				1346	3.4	1411 m	1411 m	1377 vw	1416	171.5	1408 s	1406 m		1415	156.1	1408 s	1402 m		1406 s	
	ν_s_COO; δCHOH	1402 vs	1402 s	1402 w	1397	178.2	1360 m	1359 s	1360 vw	1394	244.4	1392 s	1389 m		1346	1.9	1392 s	1365 s	1364 w	1391 s	1398 m
	τCHOH	1251 m	1252 vw	1254 w	1306	1.6	1258 w	1257 w	1256 vw	1305	1.8	1262 m	1260 w		1306	1.6	1262 m			1262 m	
	ωCHOH				1246	3.7	1224 vw	1224 vw	1225 vw	1242	4.2				1240	5.2					
9a	β(CH)	1194 s	1194 w	1195 m	1204	13.3	1190 m	1190 m	1184 m	1204	17.3	1190 m	1190 w	1190 w	1205	21.9	1191 m	1201 m	1201 m	1191 s	1200 s
9b	β(CH)			1165 m	1190	27.9	1164 vw	1164 vw	1162 w	1188	26.0			1177 m	1189	25.1			1178 w		
	β(CH)	1082 vs	1082 m				1085 m	1085 m				1089 m	1091 w				1089 m	1083 w		1089 m	1081 s
18a	β(CH)	1066 vs	1066 s	1066 w	1084	54.3	1065 m	1065 s	1066 vw	1082	57.2	1061 s	1061 m	1060 vw	1080	55.5	1062 s	1057 s	1056 w	1061 s	1054 s
18b	β(CH)	1028 m	1028 vw	1030 m	1120	39.7	1025 w	1025 w	1027 m	1119	38.3	1028 m	1028 w	1029 w	1119	32.3	1028 m	1030 w	1031 w	1028 m	1031 w
		1003 m	1003 w	1002 vs				1003 vw	1002 s					1003 s			1003 vw	1003 vw	1003 s		
5	γ(CH)	950 s	952 m	953 vw	1000	0.5	932 w	932 m		998	0.6	935 w	935 vw	935 w	995	0.2	935 w	930 m	935 vw	936 m	928 m
10a	γ(CH)	915 m	915 vw		858	0.3	904 vw	905 vw	907 vw	858	0.4	909 w	909 vw	907 w	857	0.4	911 w	901 w	901 w	909 w	
	β_s_COO	868 m	868 w	874 m	803	21.6	868 w	868 w	871 m	795	12.9	870 vw		869 w	789	15.2	870 vw	860 w	875 w	870 vw	
11	γ(CH)	796 m	795 m	795 w	744	70.33	782 w	781 w	782 m	742	54.9	789 w	787 w	788 w	741	54.8	789 w	786 m	787 w	787 m	785 m
4	ϕ(CC)	744 vs	742 m	746 w	709	23.33	728 m	727 vs	732 vw	709	28.8	736 m	735 m		709	32.2	736 m	746 s	748 w	736 m	745 s
1	α(CCC)	695 vs	695 s		634	4.84	693 m	691 vs		634	1.1	695 s	695 m		633	1.1	695 s	703 s		697 s	702 m
	γ_s_COO^–^	626 s	627 m	618 m	620	53.9	622 w	620 m	618 m	616	14.6	625 m	623 w	618 w	614	15.0	626 m		619 w	625 m	
	γOH				589	107.2				539	54.0	606 m			558	75.0	608 w	606 m		608 m	603 m
	β_as_COO	514 m		510 w	527	36.8	507 m		511 vw	498	53.0	524 m			515	24.4	525 m			525 m	531 m
16b	ϕ(CC)	450 s			469	31.8			483 vw	489	3.0	494 w		479 vw	485	4.0					
16a	ϕ(CC)	430 s			413	1.6				414	0.3				413	0.2					

s—strong; m—medium; w—weak; v—very; sh—shoulder; the symbol “ν” denotes stretching vibrations; “β” denotes in-plane bending modes; “γ” designates out-of-plane bending modes; “φ(CCC)” denotes the aromatic ring out-of-plane bending modes; “α(CCC)” designates the aromatic ring in-plane bending modes; “τ” denotes twisting out-of-plane bending; “ω” denotes wagging out-of-plane bending; “δ” denotes scissoring in-plane bending.

**Table 2 ijms-24-03078-t002:** ^1^H chemical shift for mandelic acid and its alkali metal salts (in experimental ^1^HNMR spectra and theoretically calculated in B3LYP/6-311++G(d,p)).

	Mandelic Acid		Alkaline Metal Salts of Mandelic Acid									
			Lithium		Sodium		Potassium		Rubidium		Cesium	
	Calc.	Exp.	Calc.	Exp.	Calc.	Exp.	Calc.	Exp.	Calc.	Exp.	Calc.	Exp.
H2	7.89	7.45	7.99	7.38	8.06	7.39	8.12	7.40	-	7.37	-	7.35
H3	7.69	7.35	7.56	7.23	7.52	7.23	7.55	7.22	-	7.26	-	7.20
H4	7.62	7.34	7.53	7.17	7.44	7.17	7.46	7.14	-	7.20	-	7.13
H5	7.61	7.35	7.59	7.23	7.55	7.23	7.54	7.22	-	7.26	-	7.20
H6	7.75	7.45	8.07	7.38	8.06	7.39	8.08	7.40	-	7.37	-	7.35
H7a	5.47	5.08	5.27	4.54	5.23	5.12	5.08	4.52	-	4.71		4.40
H7b	3.37	12.69	3.52	4.54	3.60	4.55	3.81	4.42	-	4.71	-	4.40
H8	6.60	12.69	-	-	-	-	-	-	-	-	-	-

**Table 3 ijms-24-03078-t003:** ^13^C chemical shifts for mandelic acid and its alkali metal salts (in experimental ^13^C NMR spectra and theoretically calculated using B3LYP/6-311++G(d,p)).

Atom No.	Mandelic Acid		Alkaline Metal Salt of Mandelic Acid									
			Lithium		Sodium		Potassium		Rubidium		Cesium	
	Calc.	Exp.	Calc.	Exp.	Calc.	Exp.	Calc.	Exp.	Calc.	Exp.	Calc.	Exp.
C1	146.02	140.39	148.97	143.38	150.81	143.86	150.69	144.18	-	142.39	-	144.37
C2	129.71	126.85	132.20	126.01	131.86	125.98	132.65	125.90	-	126.43	-	125.69
C3	133.36	128.34	132.17	126.40	132.08	126.32	131.69	126.34	-	126.60	-	126.23
C4	133.39	127.87	131.32	127.29	130.50	127.79	130.67	127.30	-	127.59	-	127.16
C5	133.15	128.34	132.52	126.40	132.42	126.32	131.87	126.34	-	126.60	-	126.23
C6	133.58	126.85	127.83	126.01	127.47	125.98	127.68	125.90	-	126.43	-	125.69
C7	75.56	72.63	76.33	73.83	76.63	73.63	76.46	73.74	-	72.99	-	73.57
C8	182.50	174.36	194.80	173.95	189.57	174.85	188.59	174.52	-	177.92	-	173.44

**Table 4 ijms-24-03078-t004:** Aromaticity indices for mandelic acid and lithium, sodium, and potassium mandelate.

Aromaticity Indices	MA	MA–Li	MA–Na	MA–K
ΔCC ^a^	0.0037	0.0069	0.0074	0.0077
ΔCCC ^b^	0.9950	1.194	1.401	1.573
HOMA	0.989	0.986	0.984	0.983
GEO	0.001	0.002	0.002	0.002
EN	0.010	0.012	0.014	0.015
I6	98.74	98.16	97.85	97.69
NICS	−8.15	−7.98	−8.02	−8.01

^a^—the difference between the length of the longest and shortest bond in an aromatic ring; ^b^—the difference between the largest and smallest angle in an aromatic ring.

**Table 5 ijms-24-03078-t005:** NBO analysis for mandelic acid and its salts with lithium, sodium, and potassium (calculated in DFT).

Atom	MA	MA–Li	MA–Na	MA–K
C1	−0.060	−0.052	−0.048	−0.045
C2	−0.192	−0.197	−0.197	−0.197
C3	−0.197	−0.201	−0.204	−0.206
C4	−0.200	−0.210	−0.214	−0.217
C5	−0.194	−0.199	−0.202	−0.203
C6	−0.190	−0.199	−0.201	−0.202
C7	0.036	0.037	0.036	0.032
C8	0.799	0.767	0.767	0.770
H8/M	0.488	0.938	0.939	0.959
H2	0.207	0.216	0.217	0.217
H3	0.205	0.201	0.199	0.198
H4	0.205	0.202	0.200	0.199
H5	0.206	0.201	0.200	0.198
H6	0.220	0.224	0.223	0.223
H7a	0.201	0.198	0.193	0.189
H7b	0.481	0.484	0.485	0.485
O1	−0.729	−0.744	−0.751	−0.756
O2	−0.671	−0.817	−0.804	−0.806
O3	−0.613	−0.848	−0.837	−0.838

**Table 6 ijms-24-03078-t006:** Values of selected reactivity parameters (descriptors) calculated for ligand (MA) and salts/complexes with selected alkali metals.

Descriptor	MA II	MA–Li	MA–Na	MA–K
Energy _HOMO_ [eV]	−7.208	−6.536	−6.275	−6.076
Energy _LUMO_ [eV]	−1.181	−1.301	−1.820	−1.502
Δ = LUMO–HOMO [eV]	6.027	5.236	4.455	4.574
Electron affinity [eV]	1.181	1.301	1.820	1.502
Ionization potential [eV]	7.208	6.536	6.275	6.076
Chemical Hardness [eV]	3.014	2.618	2.227	2.287
Chemical Softness [eV]	0.166	0.191	0.224	0.219
Electronegativity [eV]	4.195	3.919	4.048	3.789
Electrophilicity [eV]	2.919	2.933	3.678	3.139
Nucleophilicity [eV]	−7.208	−6.536	−6.275	−6.076

**Table 7 ijms-24-03078-t007:** The MIC and MBC of mandelic acid against tested microorganisms.

Microorganism	MIC (mg/mL)	MBC/MFC (mg/mL)
*Listeria monocytogenes* ATCC 13932	1.50	2.00
*Staphylococcus aureus* ATCC 25923	1.75	2.00
*Escherichia coli* ATCC 25922	2.00	2.25
*Salmonella typhimurium* ATCC 14028	2.25	2.25
*Bacillus subtilis* ATCC 6633	1.50	>5.00
*Loigolactobacillus backii* KKP 3566	>5.00	>5.00
*Candida albicans* ATCC 10231	>5.00	>5.00
*Rhodoturula mucilaginosa* KKP 3560	>5.00	>5.00

**Table 8 ijms-24-03078-t008:** The effect of mandelic acid on the maximum growth rate (µ_max_) of the tested strains—Gompertz analysis.

The Concentration of Mandelic Acid (mg/mL)	µmax [h^−1^]							
	*L. monocytogenes* ATCC 13932	*S. aureus* ATCC 25923	*E. coli* ATCC 25922	*S. typhimurium* ATCC 14028	*B. subtilis* ATCC 6633	*L.backii* KKP 3566	*C. albicans* ATCC 10231	*R. mucilaginosa* KKP 3560
0.00 (strain control)	0.058 ± 0.006 ^bC^	0.099 ± 0.003 ^dE^	0.074 ± 0.002 ^bD^	0.073 ± 0.006 ^cCD^	0.034 ± 0.008 ^cB^	0.037 ± 0.001 ^eB^	0.137 ± 0.000 ^fF^	0.016 ± 0.002 ^cA^
0.75	0.017 ± 0.000 ^aB^	0.054 ± 0.000^cD^	0.070±0.001 ^bE^	0.060 ± 0.005 ^cD^	0.007 ± 0.001 ^aA^	0.028 ± 0.001 ^dC^	0.137 ± 0.002 ^fF^	0.012 ± 0.004 ^bAB^
1.25	0.017 ± 0.002 ^aB^	0.010 ± 0.001 ^bA^	0.064 ± 0.007 ^bE^	0.042 ± 0.009 ^bD^	0.009 ± 0.001 ^bA^	0.026 ± 0.001 ^cdC^	0.136 ± 0.001 ^fF^	0.011 ± 0.003 ^bA^
1.50	TI	0.007 ± 0.001 ^abA^	0.034 ± 0.010 ^aC^	0.019 ± 0.006 ^aBC^	TI	0.025 ± 0.003 ^bcd BC^	0.130 ± 0.001 ^deD^	0.011 ± 0.003 ^bB^
1.75	TI	TI	0.031±0.011 ^aA^	0.016 ± 0.004 ^aC^	TI	0.025 ± 0.003 ^bcdBC^	0.133 ± 0.001 ^efD^	0.011 ± 0.000 ^bB^
2.00	TI	TI	TI	0.012 ± 0.001 ^aA^	TI	0.025 ± 0.003 ^bcdB^	0.127 ± 0.002 ^dC^	0.011 ± 0.000 ^bA^
2.25	TI	TI	TI	TI	TI	0.020 ± 0.001 ^bcB^	0.126 ± 0.001 ^dC^	0.011 ± 0.000 ^bA^
2.50	TI	TI	TI	TI	TI	0.020 ± 0.001 ^bcB^	0.120 ± 0.001 ^cC^	0.010 ± 0.000 ^bA^
3.00	TI	TI	TI	TI	TI	0.019 ± 0.001 ^bB^	0.109 ± 0.002 ^bC^	0.001 ± 0.000 ^bA^
5.00	TI	TI	TI	TI	TI	0.011 ± 0.001 ^aB^	0.058 ± 0.001 ^bA^	0.004 ± 0.000 ^aA^

All data were the mean ± SD, n = 3; a–f values denoted with the same letter are significantly different (*p* < 0.05) for results obtained for a single strain at different MA concentrations. A–F values denoted with the same letter are significantly different (*p* < 0.05) for results obtained for all strains at the same MA concentration. TI—total inhibition.

**Table 9 ijms-24-03078-t009:** The type of medium and growth conditions used for tested strains.

	Strain	Medium	Incubation Condition
Gram-positive	*Listeria monocytogenes* ATCC 13932	MH broth/agar	37 °C/24 h
	*Staphylococcus aureus* ATCC 25923		
	*Bacillus subtilis* ATCC 6633	MRS broth/agar	30 °C/72 h
	*Loigolactobacillus backii* KKP 3566		
Gram-negative	*Escherichia coli* ATCC 25922	MH broth/agar	37 °C/24 h
	*Salmonella* Typhimurium ATCC 14028		
Yeast	*Candida albicans* ATCC 10231	YPD broth/agar	25 °C/72 h
	*Rhodotorulla mucilaginosa* KKP 3560		

## Data Availability

Not applicable.

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
