# Peer review of "Research on the Electron Structure and Antimicrobial Properties of Mandelic Acid and Its Alkali Metal Salts"

_ijms, 2023, doi:10.3390/ijms24043078_

Round 1

Reviewer 1 Report

I have read the Manuscript " Research on the electron structure and antimicrobial properties of mandelic acid and its alkali metal salts” by Renata ÅšwisÅ‚ocka et al. The work is a combined experimental and theoretical investigation on the structure, spectroscopic and antimicrobial properties of mandelic acid and its alkali metal salts. In detail, the authors performed experimental characterization by means of FTIR, Raman and NMR spectroscopies and DFT calculations by using the hybrid functional B3LYP and 6-311++G(d,p) as basis set.

Furthermore, the authors tested antimicrobial activities of the considered compounds against 6 different bacteria strains.

The work is interesting, the results are sound and the conclusions are well supported by the achieved information. In my opinion, only minor stylist corrections and a revision of the English language are needed before the manuscript can be accepted for publication.

Just to mention a few things to be corrected:

1)      Figure legends and tables are not all easily readable. E.g. tables 1 and 8 and figures 5 and 6

2)      Some section titles should be reformulated. E.g. Theoretical calculated; NBO; HOMO/LUMO etc.

Author Response

Just to mention a few things to be corrected:

1)      Figure legends and tables are not all easily readable. E.g. tables 1 and 8 and figures 5 and 6

 Answer: Tables and figures has been correected

2)      Some section titles should be reformulated. E.g. Theoretical calculated; NBO; HOMO/LUMO etc.

Answer: The titles of some chapters have been corrected

In addition, author's corrections were made. Changes in the text are marked in red (deletions and added/corrected texts). English grammar checked and corrected.

Reviewer 2 Report

The authors aimed to investigate the structural and spectroscopic properties of mandelic acid (MA) and its alkali metal salts (from Li to Cs) along with their antimicrobial study. For this, they applied various experimental (FT-IR ATR and in KBr pellets, 1H and 13C NMR, Raman) and quantum chemical computations. The antimicrobial activity was tested against 6 bacteria and two yeasts strains. The conclusions are supported by the presented results therefore I recommend the work for publication after a major revision.

There are a few major issues that need to be dealt with before acceptance in my opinion:

1. Material and methods (now Section 3) should be moved before the Results and Discussion (now Section 2). It would make more sense to me as many of the terms are defined there (e.g. NBO, OD, MIC, MBC, etc.). Moreover, one should learn first the details of the experimental and theoretical studies, for example what solvent was used for the NMR studies, what was the level of theory for the computation, etc. Obviously, one could simply go to the respective page before reading the discussion section, but it still seems rather odd to me to put the materials and methods section inbetween the discussion and the conclusion sections. 

2. Similarly, I do not see the reason why section 2.1 (IR and Raman spectroscopy) precedes the discussion of the theoretical results (now section 2.2). This would ensure that the reader could see the structure of MA before the spectroscopic discussion. Related to this, the readers would certainly welcome a Figure similar to Figure 2 visualizing the optimized geometry of the alkali metal salts. 

3. This one is the most important issue in my opinion. The FT-IR spectra in Figure 1 seems to contradict with the assignments in Table 1. First, it was rather difficult to decipher which spectra belongs to which species since the caption states that the uppermost trace belongs to MA, while I am almost sure that in reality it is the bottom one. I think it would be a better way to label each traces with bold letters and define them in the captions. Anyway, my principal problem is with the bands assigned to the OH stretching vibrations. What I can see from the spectra is that there are at least two OH groups that feel a different environment. What I mean is that there are two bands in the particular region: one sharp and mostly intense and one broad one. I would suspect that the former one belongs to a 'free' OH group which does not participate in any H-bonding whereas the latter one is due to OH groups that form H bonds. Regarding that the presented spectra were taken in KBr, I am afraid that the broad feature comes from the water adsorbed by the pellet (which is a known phenomenon) and only the sharp peak is caused by the investigated molecule. One could decide by having a look at the ATR spectrum: if the broad bands are absent (or much weaker) then I would suggest that they are caused by the presence of adsorbed water. Unfortunately the ATR spectra are not published, maybe they should be (at least in a supplementary material together the Raman spectra), so my only clue is the relative intensities listed in Table 1: they are usually 's' (strong) in KBr and 'w' (weak) in the ATR spectrum. This would also mean that the frequencies listed in Table 1 belonging to the OH stretching vibrations are erroneous and the position of the narrow peaks should be listed instead. One proof for that could be the computed IR frequencies; their relative position (frequency shifts) between the alkali metal salts reflect better those of the narrow peaks in the experimental spectra. One might say in turn that there is no such narrow peak in the Li salt but I think there is a minor one at around 3550 cm-1. Furthermore, interestingly, the second and third traces from the top are surprisingly similar to each other, even some listed peak positions are practically identical. Are the authors sure that they belong to different alkali salts?

4. Table 1 is also pretty difficult to interpret, I would recommend the authors to decrease the font size so that all values can fit into one row, if possible. Moreover, somehow the vibrational assignment for MA should also be included (maybe in a separate table) for the sake of completeness. Besides, are the numbers in bold in the first column denote the vibrational modes? If so, why only a part of the rows contain them? Lastly, the experimental and theoretical frequency shifts of the COO symmetric stretching vibration do not seem to conform with each other, a reassignment could be considered for this vibrational mode. I am not saying the current assignment is wrong, only encouraging the authors to make sure. 

Minor issues:

1. Lines 49–50: the word 'series' is used three times in one sentence, it should be reformulated somehow

2. Line 65: missing commas "lithium, sodium, potassium rubidium and cesium" --> "lithium, sodium, potassium, rubidium, and cesium". The same holds true in the captions of Figure 1 and Table 1.

3. 'cesium' and 'caesium' both occur in the text, only one of them should be used

4. Lines 90, 93, and 98: the numbers denoting the vibrational modes could be typed in bold as they are in Table 1. 

5. Caption of Figure 1: '(upper curves)' --> '(bottom curve)'

6. Subsection '2.1.1.1. H NMR' --> '2.2.1. 1H NMR'

7. Line 122: 'aromatic ring i mandelic acid' --> 'aromatic ring in mandelic acid'

8. Sentence in lines 123–125: Can it be explained somehow why H7a is absent from the spectrum as it has a unique shift therefore it is expected to be easily distinguished?

9. Sentence in lines 126–127: Is there a reason for the huge discrepancy between the theoretical and experimental shifts of the H8 carboxylic proton? Is it because the insufficient computational level, the formation of H bonded dimers, or something else?

10. Sentence in lines 129–131: Can the authors explain why the theoretical values does not seem to be consistent with the trend found for the experimental ones? Can it be due to the solvent model used or something else?

11. Subsection '2.1.1.13. CNMR' --> '2.2.2. 13C NMR'

12. Lines 148–149: "The carbons chemical shifts of the aromatic ring" --> "The chemical shifts of the carbon atoms in the aromatic ring"

13. Section '2.2. Theoretical calculated' --> '2.2. Theoretical computations'

14. Line 186: Caption of Table 4 remained in Polish, please translate it into English :)

15. Line 191: "structure with greater stability" --> "structure with the largest stability"

16. Line 195: 'Table 5' --> 'Table 4'

17. Sentences in lines 205–210: similarly to point 10, can the authors explain the difference between the tendencies for the theoretical and experimental NMR shifts?

18. Page 10, Figure 3, Table 6, and Line 232: both 'Delta' and 'DeltaE' are used for the LUMO–HOMO energy difference, please make it uniform.

19. Heading of Table 6: 'MA (2)' --> 'MA (II)' I suppose the number here represents conformer II, which is denoted with the Roman numeral II throughout the manuscript except here.

20. Table 8 also seems cluttered, I hope it could be improved somehow.

21. Line 309: 'beyond B. subtilis' – which one does this phrase refer to?

22. Line 309: 'for this strain' – which one exactly, I am not sure?

23. Sentence in lines 327–328: the statement is not true for E. coli, am I right?

24. Sentence in lines 331–333: the statement is not true for S. aureus if I am not mistaken.

25. Subsection '3.1. Theoretical calculated' should be renamed to '3.1. Theoretical computations'

Author Response

  1. Material and methods (now Section 3) should be moved before the Results and Discussion (now Section 2). It would make more sense to me as many of the terms are defined there (e.g. NBO, OD, MIC, MBC, etc.). Moreover, one should learn first the details of the experimental and theoretical studies, for example what solvent was used for the NMR studies, what was the level of theory for the computation, etc. Obviously, one could simply go to the respective page before reading the discussion section, but it still seems rather odd to me to put the materials and methods section inbetween the discussion and the conclusion sections.

Answer: The order of the sections results from the requirements of the journal, which is why the article was prepared in this way.

  1. Similarly, I do not see the reason why section 2.1 (IR and Raman spectroscopy) precedes the discussion of the theoretical results (now section 2.2). This would ensure that the reader could see the structure of MA before the spectroscopic discussion. Related to this, the readers would certainly welcome a Figure similar to Figure 2 visualizing the optimized geometry of the alkali metal salts.

Answer:

The authors concluded that experimental research plays an important role in the work, while theoretical calculations are a data supplement. That is why the individual sections have been planned in this way.

  1. This one is the most important issue in my opinion. The FT-IR spectra in Figure 1 seems to contradict with the assignments in Table 1. First, it was rather difficult to decipher which spectra belongs to which species since the caption states that the uppermost trace belongs to MA, while I am almost sure that in reality it is the bottom one. I think it would be a better way to label each traces with bold letters and define them in the captions. Anyway, my principal problem is with the bands assigned to the OH stretching vibrations. What I can see from the spectra is that there are at least two OH groups that feel a different environment. What I mean is that there are two bands in the particular region: one sharp and mostly intense and one broad one. I would suspect that the former one belongs to a 'free' OH group which does not participate in any H-bonding whereas the latter one is due to OH groups that form H bonds. Regarding that the presented spectra were taken in KBr, I am afraid that the broad feature comes from the water adsorbed by the pellet (which is a known phenomenon) and only the sharp peak is caused by the investigated molecule. One could decide by having a look at the ATR spectrum: if the broad bands are absent (or much weaker) then I would suggest that they are caused by the presence of adsorbed water. Unfortunately the ATR spectra are not published, maybe they should be (at least in a supplementary material together the Raman spectra), so my only clue is the relative intensities listed in Table 1: they are usually 's' (strong) in KBr and 'w' (weak) in the ATR spectrum. This would also mean that the frequencies listed in Table 1 belonging to the OH stretching vibrations are erroneous and the position of the narrow peaks should be listed instead. One proof for that could be the computed IR frequencies; their relative position (frequency shifts) between the alkali metal salts reflect better those of the narrow peaks in the experimental spectra. One might say in turn that there is no such narrow peak in the Li salt but I think there is a minor one at around 3550 cm-1. Furthermore, interestingly, the second and third traces from the top are surprisingly similar to each other, even some listed peak positions are practically identical. Are the authors sure that they belong to different alkali salts?

Answer:

Table 1 and figure 1 were corrected and supplemented according to the reviewer's suggestions. In particular, OH vibrations were analyzed. The IR spectra of KBR and ATR were compared.

  1. Table 1 is also pretty difficult to interpret, I would recommend the authors to decrease the font size so that all values can fit into one row, if possible. Moreover, somehow the vibrational assignment for MA should also be included (maybe in a separate table) for the sake of completeness. Besides, are the numbers in bold in the first column denote the vibrational modes? If so, why only a part of the rows contain them? Lastly, the experimental and theoretical frequency shifts of the COO symmetric stretching vibration do not seem to conform with each other, a reassignment could be considered for this vibrational mode. I am not saying the current assignment is wrong, only encouraging the authors to make sure. 

Answer:

The table 1 has been corrected. Column 1 shows vibration designations according to Versanyi (citation inserted). The spectrum of mandeic acid along with the assignment and detailed interpretation were published by the authors in an earlier work. The correctness of the carboxylate anion vibration analysis was checked.

Minor corrections (1-25)

Has been corrected.

In addition, author's corrections were made. Changes in the text are marked in red (deletions and added/corrected texts). English grammar checked and corrected.

Reviewer 3 Report

The MS is interesting since it represents analyses of the structure, spectroscopic (FTIR, Raman, NMR) and antimicrobial properties of mandelic acid (MA) as well as its alkali metal salts. All spectra are calculated by DFT on the ground of geometry optimization.

Antimicrobial activities of MA and its salts were tested against 6 bacteria including Gram-positive Listeria monocytogenes, Staphylococcus aureus, Gram-negative Escherichia coli and Salmonella, two yeasts and Aspergillus niger. The aromaticity of MA ligand, under the salt formation, decreases in the series with the increase of the metal ionic potential and the reactivity of the compounds increases with the destabilization of the aromatic MA system. Meanwhile, the aromaticity analysis seems to be rather artificial without connection with real biochemical mechanisms of antimicrobial activities. The same concerns the spectral studies, which have their own significance, but no connection with antimicrobial action. In general the MS requires revision and accounts for following items.

1) nothing is said about the scaling factor in IR frequency calculations by DFT method. Table 1 is bad because of the shifts in columns. The most important modes should be shown in table (together with MA modes as standard) and the big table for the whole spectral assignments could be in Supplementary.

2) what does it mean ATR? Why there is no MA assignment in IR and Raman?

3) Three structures of mandelic acid conformers were optimized by the B3LYP/6-311++g(d,p) method. "Figure 2 shows the optimized structures I-III and the monomer structure solved by X-ray diffraction [30]. Conformer II has the lowest energy. This structure is identical to the real structure of mandelic acid according to x-ray diffraction data (Fig.2.)" Perfect! Why more quantity details are not shown? Direct comparison of XRD data with DFT prediction would be much more interesting than the abstract aromaticity indexes.

4) The shape and position of the HOMO and LUMO orbitals are determined  in Fig.3. The HOMO-LUMO excitation provides the charge transfer state for all salts. Why there are no any indication of this CT absorption in UV-vis spectra? Why there is no any discussion of CT influence on reactivity?

Some important references are missed. About scaling factor: Vibrational spectra of the steroid hormones, estradiol and estriol, calculated by density functional theory. The role of low-frequency vibrations

DN Hovorun et al. Ukr. Biokhim. Zh 80 (4), 82-95 (2008)

About Electrophilicity: State-dependent global and local electrophilicity of the aryl cations

SV Bondarchuk, et al.

State-dependent global and local electrophilicity of the aryl cations

The Journal of Physical Chemistry A 118 (17), 3201-3210.

The MS needs major revision

Author Response

1) nothing is said about the scaling factor in IR frequency calculations by DFT method. Table 1 is bad because of the shifts in columns. The most important modes should be shown in table (together with MA modes as standard) and the big table for the whole spectral assignments could be in Supplementary.

Answer: Table 1 has been revised for greater clarity. The location of the bands in the spectra of the studied compounds is quite an important issue regarding the electronic structure, which is why the authors have included detailed data in the main part of the work. For calculated IR spectra applied Vibrational frequency scaling factors IRfactors=0.98. This information has been supplemented in the methodology.

2) what does it mean ATR? Why there is no MA assignment in IR and Raman?

Added the text: The analysis of infrared spectra made in the KBr matrix and the ATR reflection technique showed that the use of the KBr matrix had no significant effect on the band shifts in the spectra of the tested compounds.

3) Three structures of mandelic acid conformers were optimized by the B3LYP/6-311++g(d,p) method. "Figure 2 shows the optimized structures I-III and the monomer structure solved by X-ray diffraction [30]. Conformer II has the lowest energy. This structure is identical to the real structure of mandelic acid according to x-ray diffraction data (Fig.2.)" Perfect! Why more quantity details are not shown? Direct comparison of XRD data with DFT prediction would be much more interesting than the abstract aromaticity indexes.

Answer: The purpose of calculating mandelic acid conformers was to show that the structure with the lowest energy is the same as the real structure. The aromaticity indices provided an additional tool to evaluation  the effect of salt formation on the electron system of the ligand.

4) The shape and position of the HOMO and LUMO orbitals are determined  in Fig.3. The HOMO-LUMO excitation provides the charge transfer state for all salts. Why there are no any indication of this CT absorption in UV-vis spectra? Why there is no any discussion of CT influence on reactivity?

Answer:

UV-VIS spectra were not recorded in this work, therefore no such discussion was undertaken. Based on the study of alkali metal salts with many complexes such as aromatic acids, we can conclude that the bands associated with CT transitions were not observed in the UV-VIS spectra.

In addition, author's corrections were made. Changes in the text are marked in red (deletions and added/corrected texts). English grammar checked and corrected.

Round 2

Reviewer 2 Report

The authors mostly addressed my concerns. Nevertheless, two typos have been accidentally introduced during the revision process:

Line 213, Table 4 title: 'Aromaticiti' --> 'Aromaticity'

Line 243, subsection title: 'reactiviti' --> 'reactivity'

Author Response

Line 213, Table 4 title: 'Aromaticiti' --> 'Aromaticity'

Line 243, subsection title: 'reactiviti' --> 'reactivity'

Answer: Has been changed.

Reviewer 3 Report

The MS "Research on the electron structure and antimicrobial properties of mandelic acid and its alkali metal salts" (ijms-2177806)
is improved after revision but still suffers many errors. The MS is interesting and can be published after new revision, but the authors ignore some reviewer's comments. Various aspects of theoretical quantum chemistry calculations are described in a very fragmented manner. The whole final MS reduction is floppy prepared.

I am sure that not many readers are familiar with the attenuated total reflection (ATR) technique (even those who use FTIR KBr tablets). 

That is why one can read in the row 116 page 3: "ATR reflection". The whole phrase should be like this: The analysis of infrared spectra detected in the KBr matrix and with the ATR technique. 

The new abzaz in Page 2 (rows76-83) is rather thrivial and needs reduction. In the last sentence clarify the points. 

What does it means "Int." in Table 1? Is it IR absorption intensity (in what units?) or is it Raman scattering activity (A4/a.m.u.) or Raman intensity (recalculated from DFT output with the exciting laser wavelength account)? The comments to Table 1 are necessary for reader. Table 1 is a result of great work and needs more careful description.

Page 4, row 421:  "Calculations were made using the B3LYP/6-311++G(d,p) density functional method" (DFT) Basis set repeated 3 times here. The word DFT should be denoted and used in the main text (instead of B3LYP/6-311++G(d,p) abbreviation) which is mentioned 13 times in the text.

Why "the HOMO and LUMO orbitals and the reactivity descriptors were calculated using the HF/6-311++G(d,p) method"? (row 419).  Why not from the same DFT B3LYP? The Hartree-Fock as the SCF MO method or just a proceedure in self-consistent DFT?

HOMA and JULG aromaticity indices are still misterious without any references and explanations. The Harmonic Oscillator Model of Aromaticity is a rather primitive index in comparison with modern magnetic indexes (NICS  nucleus independent chem. shift, GIMIC, etc). If authors concentrate on aromaticity (which is not very relevant to the whole study) so much let them study it at the modern proper level. I still recommend to shift it to supplimemntary since HOMA and JULG have never used in discussions of antimicrobial propertiesPage 3, row 222: "This is confirmed by the decreasing values of HOMA and JULG aromaticity indices. in 222 the same row". (Point). The final reduction of the MS includes a number of errors.

The MS requiers new revision

Author Response

  • I am sure that not many readers are familiar with the attenuated total reflection (ATR) technique (even those who use FTIR KBr tablets). 

The new abzaz in Page 2 (rows76-83) is rather thrivial and needs reduction. In the last sentence clarify the points. 

Answer: Added text :

The spectra were made using two techniques of infrared spectroscopy, i.e. by pressing the sample with the matrix compound KBr and the ATR technique (attenuated total reflection on the surface). Both techniques allow to obtain similar results, however, using the apparatus equipped with a zinc selenide crystal, we obtained ATR spectra, in which not all signals were clear (the spectra are included in the supplement). On the other hand, the comparison of the FTIR spectra of KBr and ATR made it easier to determine of the bands originating from the vibrations of the hydroxyl group.

That is why one can read in the row 116 page 3: "ATR reflection". The whole phrase should be like this: The analysis of infrared spectra detected in the KBr matrix and with the ATR technique. 

Answer: Has been changed.

  • What does it means "Int." in Table 1? Is it IR absorption intensity (in what units?) or is it Raman scattering activity (A4/a.m.u.) or Raman intensity (recalculated from DFT output with the exciting laser wavelength account)? The comments to Table 1 are necessary for reader. Table 1 is a result of great work and needs more careful description.

Answer: unit added for the intensity [KM/Mole]

  • Page 4, row 421:  "Calculations were made using the B3LYP/6-311++G(d,p) density functional method" (DFT) Basis set repeated 3 times here. The word DFT should be denoted and used in the main text (instead of B3LYP/6-311++G(d,p) abbreviation) which is mentioned 13 times in the text.

Answer: Has been changed.

  • Why "the HOMO and LUMO orbitals and the reactivity descriptors were calculated using the HF/6-311++G(d,p) method"? (row 419).  Why not from the same DFT B3LYP? The Hartree-Fock as the SCF MO method or just a proceedure in self-consistent DFT?

Answer: Has been fixed on DFT (mistake in text)

  • HOMA and JULG aromaticity indices are still misterious without any references and explanations. The Harmonic Oscillator Model of Aromaticity is a rather primitive index in comparison with modern magnetic indexes (NICS  nucleus independent chem. shift, GIMIC, etc). If authors concentrate on aromaticity (which is not very relevant to the whole study) so much let them study it at the modern proper level. I still recommend to shift it to supplimemntary since HOMA and JULG have never used in discussions of antimicrobial propertiesPage 3, row 222: "This is confirmed by the decreasing values of HOMA and JULG aromaticity indices. in 222 the same row". (Point). The final reduction of the MS includes a number of errors.

Answer:

The chapter on aromaticity has been supplemented. Additionally, the Bird and NICS indexes were calculated. There is much controversy over the use of aromaticity indices. In the systems we studied, these indices perform quite well. we have used them many times. The NICS index actually performed well as well. We have not used other indices like FLU, GIMIC, but we will consider it in the next  works. However, calculating indexes is not our primary goal, we only use it in addition. Enough spectroscopic tools show the effects we are studying. Indexes are just an add-on.

Added ATR and Raman spectra to the supplement.